# Genetic and Proteomic Basis of Sclerotinia Stem Rot Resistance in Indian Mustard [*Brassica juncea* (L.) Czern & Coss.]

**DOI:** 10.3390/genes12111784

**Published:** 2021-11-10

**Authors:** Manjeet Singh, Ram Avtar, Nita Lakra, Ekta Hooda, Vivek K. Singh, Mahavir Bishnoi, Nisha Kumari, Rakesh Punia, Neeraj Kumar, Raju Ram Choudhary

**Affiliations:** 1Oilseeds Section, Department of Genetics and Plant Breeding, CCS Haryana Agricultural University, Hisar 125 004, India; ramavtar0706@gmail.com (R.A.); vks.slay@gmail.com (V.K.S.); mahaveer.bishnoi9@gmail.com (M.B.); nishaahlawat211@gmail.com (N.K.); punia.rakesh98@gmail.com (R.P.); neerajkummar8@gmail.com (N.K.); rajuramchoudhary33@gmail.com (R.R.C.); 2Department of Molecular Biology, Biotechnology and Bioinformatics, CCS Haryana Agricultural University, Hisar 125 004, India; 3Department of Mathematics and Statistics, CCS Haryana Agricultural University, Hisar 125 004, India; ektahooda@gmail.com

**Keywords:** Sclerotinia stem rot, Indian mustard, pathogen resistance, generation mean analysis, PPIase, protein

## Abstract

Sclerotinia stem rot is one of the utmost important disease of mustard, causing considerable losses in seed yield and oil quality. The study of the genetic and proteomic basis of resistance to this disease is imperative for its effective utilization in developing resistant cultivars. Therefore, the genetic pattern of Sclerotinia stem rot resistance in Indian mustard was studied using six generations (P_1_, P_2_, F_1_, F_2_, BC_1_P_1_, and BC_1_P_2_) developed from the crossing of one resistant (RH 1222-28) and two susceptible (EC 766300 and EC 766123) genotypes. Genetic analysis revealed that resistance was governed by duplicate epistasis. Comparative proteome analysis of resistant and susceptible genotypes indicated that peptidyl-prolyl cis-trans isomerase (A0A078IDN6 PPIase) showed high expression in resistant genotype at the early infection stage while its expression was delayed in susceptible genotypes. This study provides important insight to mustard breeders for designing effective breeding programs to develop resistant cultivars against this devastating disease.

## 1. Introduction

India is the fourth largest producer of oilseeds just after the USA, China, and Brazil, accounting for about 19% of the global area and 2.7% of global production [1]. Presently, India needs 25 million tons (MT) of vegetable oils, of which merely 10.5 MT is produced domestically. Owing to this, India is also the world’s largest consumer and importer of vegetable oils and meets up to 60% of its domestic demand through imports, costing it up to USD 10 billion annually [2,3]. Moreover, the domestic edible oils demands will increase in the coming years since per capita consumption of vegetable oil is on a steady rise in India due to its ever increasing population. The low productivity of oilseed crops is the major reason behind such a huge imbalance between demand and supply of edible oils in India [4]. Among the seven edible oilseed crops grown in the country, brassica oilseeds alone contribute more than 30% to total oil production. Indian mustard [*Brassica juncea* (L.) Czern & Coss.] an allotetraploid (2n = 4x = 36, AABB), is the most predominantly cultivated crop occupying approximately 90% of the total area under brassica oilseeds cultivation in India. The average yields of brassica oilseeds are 1245 kg/hectare in India versus global productivity of 1994 kg/hectare [5,6]. Such a large instability in yield and production of this crop is mainly due to its sensitivity against various abiotic and biotic extremities which is anticipated to rise in the near future due to changing climatic conditions [7,8]. Among various stresses, fungal diseases viz., white rust, alternaria blight, downy mildew, and Sclerotinia stem rot are the major factors influencing crop productivity in Indian mustard. Among them, Sclerotinia stem rot has switched from being of minor significance to major significance since the last decade due to changes in climatic conditions and presently, one of the most devastating diseases of mustard at the global level causing up to 5–100% yield losses [8,9,10,11,12]. This disease is caused by ubiquitous, cosmopolitan soil-borne hemibiotrophic fungus, *Sclerotinia sclerotiorum* (Lib) de Bary, causing annual yield losses worth over several hundred million dollars [13].

*S. sclerotiorum* exhibits dual infection mode in its host as its resting bodies (sclerotia) and can germinate either myceliogenically (soil-borne infection) to cause disease in the basal stem or can germinate carpogenically (air-borne infection) to cause disease in leaves and siliquae (Figure 1).

Besides affecting almost all plant parts, the stem is the most affected host tissue and infection to stem is directly related to its girdling and plant lodging which is one of the most ultimate reasons for major yield losses in mustard at the field level [12,14,15].

Changing climatic conditions and modern agricultural practices increase the risk of Sclerotinia rot epidemic development by allowing the pathogen to accumulate a high inoculum load [16]. The control of this pathogen through cultural and chemical control is often very tedious and not that effective because of its complex mode of infection and its longer survival ability (up to 10 years in soil without host availability) in the form of a resting structure called sclerotia [17]. In addition, fungicide application poses a serious threat to climate and adds further cost to crop cultivation. Therefore, host genetic resistance is the most convenient, economic, and eco-friendly approach for the effective control of this devastating pathogen [18,19]. Earlier attempts to identify resistant sources against this disease in Indian mustard were hampered as all the *B. juncea* genotypes evaluated were found susceptible to Sclerotinia stem rot and any of the resistant sources reported belonged to other cruciferous crops and its wild relatives such as *Brassica napus*, *B. fruticulosa*, *B. rupestris*, *B. incana*, *B. insularis*, *B. villosa*, *Erucastrum cardaminoides*, *E. abyssinicum*, *Sinapis alba*, and *Diplotaxis tenuisiliqua* [20,21,22,23,24,25,26,27,28,29,30,31,32], with no reports available about its resistance in *B. juncea*, which is an important oil yielding crop in the Indian context. However, in recent few years, increasing attention has been paid which has ultimately led to the identification of a few Indian mustard genotypes resistant against this pathogen [3,5,33,34].

Successful infection of *S. sclerotiorum* on mustard stems leads to the development of typical symptoms in the form of white greyish, water-soaked lesions which often extend as the disease progresses and cause stem girdling which ultimately leads to lodging and wilting of plants [5]. Measurement of lesion length at a particular time after infection is generally used to assess the damage caused by this pathogen at the individual plant/stem level. This is because lesion length expansion has a direct positive relationship with disease severity, damage in the form of stem breakage, and plant collapse at the field level as well as the amount of secondary inoculum produced [15]. Therefore, measurement of stem lesion length at a particular time after infection is a very useful component for the assessment of quantitative resistance against this pathogen in Indian mustard.

The knowledge about the pattern of inheritance and the nature of gene action involved in resistance is crucial for the effective utilization of the resistant source in disease resistance breeding programs. Information regarding the genetic basis of resistance allows breeders to frame an efficient breeding strategy for the development of resistant cultivars. The generation mean analysis (GMA) is a simple but effective approach for the estimation of the nature and magnitude of gene actions (additive or dominance) involved in a particular trait. Besides this, GMA also helps breeders in the detection of various types of epistasis, viz., additive × additive, additive × dominance and dominance × dominance operating in the inheritance of a particular trait [35,36,37,38,39].

Recent advancement in plant biotechnology offers several techniques to assist crop geneticists and breeders in developing crop cultivars more efficiently. Proteomics is comparatively new tool among the different omic approaches frequently used by plant scientists. The proteome, the translational version of the genome, is a crucial functional player for mediating specific cellular processes, offering several advantages over other omics techniques. Post-translational modification reflects the functional impression of proteins at the cellular level. The information derived from proteomic studies can help plant breeders to modify plant genetic architecture and enable crop cultivars with high yield potential to improve crop qualities and various stress tolerances. The recent advancement in high-throughput analysis of crop proteins using LC-MS/MS helps to identify particular proteins. Moreover, it offers a new alternative for the determination of genes that are responsible for a particular trait. It might assist plant breeders in developing disease-resistant cultivars for sustainable agriculture [40,41,42,43]. Several recent studies indicate that the proteomic approach helps in understanding the molecular mechanism involved in plant–pathogen interaction, identification of the host’s resistance/susceptible factor(s), and pathogen’s virulence factor(s) at the molecular level [43,44,45,46].

In this context, the present investigation was designed with the aim to study the inheritance and protein expression pattern involved in Sclerotinia stem rot resistance in Indian mustard.

## 2. Materials and Methods

### 2.1. Plant Materials

The plant material comprised of six generations (P_1_, P_2_, F_1_, F_2_, BC_1_P_1_, and BC_1_P_2_) developed from crossing between Sclerotinia stem rot resistant (R) genotype, viz., RH 1222-28 [3,5,33,34] and two susceptible (S) genotypes, viz., EC 766300 and EC 766123 [3]. These six-generation population sets were designated as population-I (C-I) [RH 1222-28 (R) × EC 766300 (S)] and population-II (C-II) [RH 1222-28 (R) × EC 766123 (S)] (Table 1).

Both F_1_ crosses were attempted at Oilseeds Research Farm, Department of Genetics and Plant Breeding, CCS HAU, Hisar during *Rabi* season of 2018–19. F_1_s were selfed to obtain F_2_ population and simultaneously backcrossed to produce Backcross 1 (BC_1_P_1_) and Backcross 2 (BC_1_P_2_) generations during off-season, 2019, at national off-season nursery, Regional Station, ICAR-Indian Agricultural Research Institute (IARI), Wellington (Nilgiris), Tamil Nadu, India.

### 2.2. Crop Cultivation

All six generations of both the populations were raised in Compact Family Block Design (CFBD) with three replications during *Rabi* season, 2019–2020 (under late sown conditions), in two rows of 5 m length for non-segregating generations, viz., P_1_, P_2_, and F_1_ while three and six rows of 5 m length each for back cross (BC_1_P_1_ and BC_1_P_2_) and F_2_ population, respectively. In each replication of both of the crosses, 5 plants among the parents, 10 from F_1_, 100 from F_2_ generation, 40 from BC_1_P_1_, and BC_1_P_2_ generations were tagged for artificial stem inoculation.

### 2.3. Sclerotinia sclerotiorum Pure Culture Preparation, Artificial Stem Inoculation and Disease Assessment

The sclerotia were collected from *S. sclerotiorum* infected Indian mustard plants at permanent sick plot of Oilseeds Research Farm, Department of Genetics and Plant Breeding, CCS HAU, Hisar. These sclerotia were then properly sterilized and aseptically shifted into Petriplates containing Potato Dextrose Agar (PDA, Himedia Laboratories, Mumbai, India). These plates were then incubated at 22 ± 2 °C in a BOD incubator and sub-cultured periodically to maintain a pure culture. Five-day-old pure cultures were used for artificial stem inoculation in each tagged plant at the post-flowering stage as per the method adopted by Singh et al. [5] (Figure 2).

Lesion length (cm) from each inoculated plant was measured using a linear ruler at 20 days after inoculation. Each generation of both the populations was classified into different resistance groups based on mean lesion length (cm) as per scale given in Table 2.

### 2.4. SDS-PAGE and Sequencing of Protein

#### 2.4.1. SDS-PAGE

For SDS-PAGE, leaf tissues were ground to powder in liquid nitrogen and melted in ice-cold extraction buffer (50 mM Tris-HCl, pH 7, 0.1 mM PMSF, 0.1 mM DTT, 1 mM Ascorbic acid, 1 mM PMSF and PVP). The homogenate was centrifuged at 10,000 g for 20 min in a refrigerated centrifuge. Total soluble protein was estimated by Bradford [47] using bovine serum albumin as standard. SDS-PAGE was performed as described by Laemmli [48] in a 4% polyacrylamide (*w*/*v*) stacking gel and a 10% polyacrylamide (*w*/*v*) resolving gel. The supernatant was added with an equal volume of gel loading buffer. The mixture was heated at 100 °C for 3–5 min. The protein sample was stored at −20 °C until used for electrophoresis. Prior to loading, the stored (at −20 °C) samples were heated in a boiling water bath for 3 min. The samples were loaded in the wells. At the end of the electrophoresis, the polypeptide bands were visualized by staining with Coomassie brilliant blue R-250 (CBB). The molecular weight of the sample proteins was determined by using standard protein molecular weight markers which were run simultaneously during SDS-PAGE electrophoresis of the sample protein. The molecular weight of the unknown polypeptides was determined from their Rf value as mentioned below:$$Rf=\frac{Distance\;migrated\;by\;proteins}{Distance\;migrated\;by\;dye}$$

Further, protein band was cut from CBB stained gel and sequenced using ultra-high-resolution nano-LC MS/MS with a 10–15 Cm C18 column, Q Exactive/Lumos Series Orbitrap Based Mass Spectrometer.

#### 2.4.2. Experimental Procedure for Protein Sequencing

##### Sample Preparation

A 25 microliter sample was taken and reduced with 5 mM TCEP and further alkylated with 50 mM iodoacetamide and then digested with trypsin (1:50, trypsin/lysate ratio) for 16 h at 37 °C. Digests were cleaned using a C18 silica cartridge to remove the salt and dried using a speed vac. The dried pellet was resuspended in buffer A (5% acetonitrile, 0.1% formic acid).

##### Mass Spectrometric Analysis of Peptide Mixtures

All the experiment was performed using EASY-nLC 1200 system (Thermo Fisher Scientific, Bath, UK) coupled to Thermo Fisher-QExactive equipped with a nanoelectrospray ion source. An amount of the peptide mixture (1.0 µg) was resolved using 25 cm PicoFrit column (360 µm outer diameter, 75 µm inner diameter, 10 µm tip) filled with 1.9 µm of C18-resin (Dr Maeisch, Ammerbuch, Germany). The peptides were loaded with buffer A and eluted with a 0–40% gradient of buffer B (95% acetonitrile, 0.1% formic acid) at a flow rate of 300 nl/min for 100 min. MS data were acquired using a data-dependent top10 method dynamically choosing the most abundant precursor ions from the survey scan.

##### Data Processing

All samples were processed and RAW files generated were analyzed with Proteome Discoverer (v2.2) against the Uniprot BRASSICA reference proteome database. For Sequest and AMANDA search, the precursor and fragment mass tolerances were set at 10 ppm and 0.05 Da, respectively. The protease used to generate peptides, i.e., enzyme specificity was set for trypsin/P (cleavage at the C terminus of “K/R: unless followed by “P”) along with a maximum missed cleavages value of two. Carbamidomethyl on cysteine as fixed modification and oxidation of methionine and N-terminal were considered as variable modifications for database search. Both peptide spectrum match and protein false discovery rate were set to 0.01 FDR. Data analysis was accounted for and calculated by Proteome Discoverer software.

### 2.5. Statistical Analysis

The analysis of variance (ANOVA) for lesion length among different generations of both the populations and Duncan’s multiple range test (DMRT) for comparing means were performed using STAR version 2.0.1 Statistical Software developed by International Rice Research Institute, Manila, Philippines. Generation mean analysis (as per Hayman, [49] and Jinks and Jones [50]) by taking lesion length (cm) from every individual plant from all the six generations was performed using TNAUSTAT 2.0 version Statistical Software (TNAU, Coimbatore, India). Heritability in broad (h^2^ bs) and narrow sense (h^2^ ns), genetic advance (GA), potency ratio (PR), and effective factors/minimum number of genes were calculated as per formulae suggested by Warner [51], Johnson et al. [52]; Smith [53] and Burton [54], respectively.

## 3. Results

### 3.1. Analysis of Variance and Comparison of Mean Lesion Length

Analysis of variance showed highly significant effects of generations (*p* ≤ 0.01) on lesion length (cm) in both the populations (C-I and C-II) (Table 3). The mean lesion length comparison of different generations of both populations is presented in Table 4. Each generation, viz., parents (P_1_ and P_2_), F_1_, F_2_, and backcrosses (BC_1_P_1_ and BC_1_P_2_) of both populations (C-I and C-II) showed a range of reactions against *S. sclerotiorum*. The resistant parent RH 1222-28 had the smallest mean lesion length (Figure 3A) while the susceptible parent EC 766300 showed the highest mean lesion length (Figure 3B) followed by EC 766123 (Figure 3C).

The F_1_, F_2_, BC_1_P_1_, and BC_1_P_2_ generations in both crosses fall within the parental range except for C-II, where BC_1_P_2_ expressed a significantly higher mean lesion length than the susceptible parent EC 766123. The mean lesion length exhibited by F_1_s was always higher than their respective calculated mid-parent values which showed their skewness towards susceptible parents. F_2_ generations showed intermediate mean lesion length, viz., 8.29 and 9.57 cm for C-I and C-II, respectively, and exhibited continuous variation from highly resistant to highly susceptible plants in this generation (Figure 3D–I). The mean lesion length in the backcross generations inclined in the direction of their corresponding recurrent parents. The BC_1_P_1_ generation had a significantly higher mean lesion length than its recurrent parent while a statistically non-significant difference was observed for mean lesion length between BC_1_P_2_ and its respective recurrent parents. Based on mean lesion length, P_1_ (RH 1222-28) was found resistant while, P_2_, F_1_s, and BC_1_P_2_ were found highly susceptible. The F_2_ were found susceptible to highly susceptible while BC_1_P_1_ was susceptible.

### 3.2. Scaling Tests and Nature of Gene Action

The significance of the individual scaling test manifested the existence of epistasis and inadequacy of the simple additive-dominance model for explaining Sclerotinia stem rot resistance in both populations. This was again inveterate by significant χ^2^ values of the joint scaling test (Table 5).

In the six-parameter model, the estimates of mean, additive, and dominance effects were highly significant in both populations. On the basis of the magnitude of gene effects, the dominance (h) component was comparatively more prominent over the additive component (d). The additive × additive (i), additive × dominance (j), and dominance × dominance (l) epistasis were found to be significant for both populations. The additive (d) gene effect and the dominance × dominance (l) type of interaction was in the negative direction while all other genetic parameters were in the positive direction. The significant and opposite sign of dominance (h) gene effect as well as dominance × dominance (l) type of interaction indicated the involvement of duplicate epistasis in both populations (Table 6).

### 3.3. Genetic Parameters

The estimates of heritability, viz., broad (h^2^bs) and narrow sense (h^2^ ns) were high (>0.60) for both populations. The genetic advance (GA) was moderate, i.e., 9.33 for C-I while 8.35 for C-II. Additive variance (D) was higher than dominance (H) and environment (E) variances. The average degree of dominance ($\sqrt{D/H}$) and potency ratio (PR) estimates were less than one and greater than zero, indicating that the genes responsible for Sclerotinia stem rot resistance were partially dominant in their expression. The estimates of covariance among D and H overall loci (F) was greater than zero and in the negative direction while it was nearly zero for $F/\sqrt{H\times D}$. The effective factors/minimum number of genes conferring resistance/susceptibility ranged from 3.50 in C-I to2.46 in C-II indicating the oligogenic nature of stem rot resistance (Table 7).

### 3.4. Protein Profiling of Resistant and Susceptible Genotypes

To further understand the factors/proteins/genes responsible for variation in stem rot resistance in *Brassica*, a protein expression study was conducted. Total protein content ofleaf tissues after infection indicatedthat *S. sclerotiorum* infection leads to a decrease in total protein content as the disease progresses but it was more pronounced in susceptible genotypes. Further, an equal amount of protein extract was loaded on the SDS-PAGE for expression profiling which revealed that a total of ~20–22 proteins were detected on CBB stain gel which showed differential expression in control and pathogen-infected leaf tissues of all genotypes (RH 30, RH 1222-28, and EC 766300) selected in the present study. We observed changes in the expression of the number of peptides after the 6th, 12th and 18th day of infection in all genotypes; whereas ~18.4 kDa peptides (Figure 4A–D) were expressed exclusively during early infection stage (6th day) in resistant genotype, RH 1222-28 over susceptible genotypes (RH 30 and EC 766300) and this peptide may be denoted as pathogen-induced peptide.

However, this peptide also appeared in susceptible genotypes at 12 days after infection. It showed that this particular peptide may be responsible for resistance in RH 1222-28. The resistant genotype showed a rapid response to the pathogen at 6 days after infection, but the susceptible genotypes showed a late response to the pathogen. In order to learn more about the nature of this stress-inducible protein, the accumulated protein spot was excised from SDS-PAGE gels, and sequence analysis was performed using Nano LC-MS/MS, and homology searches were carried out using the Maskow search program. The protein has been identified as peptidyl-prolyl cis-trans isomerase based on sequencing analysis (Figure 5A,B and Appendix A).

### 3.5. Insilco Analysis of PPIase

#### 3.5.1. Phylogenetic Analysis and Domain Analysis of PPIase

The amino acid sequences of identified PPIase were compared with other homologous proteins from different species (Figure 5C) and performed multiple sequence alignments by using clustalW2. At the whole amino acid level, the identified PPIase showed 90–100% similarity with other *Brassica* species proteins and was found more closer to peptidyl-prolylcis-trans isomerase (*Brassica oleracea* var. oleracea) and (*Brassica rapa*). This shows that it is highly conserved in Brassica species. Further, identified protein sequences were analyzed for the presence of functional domain using SMART tool and found that sequences containing pro-isomerase domain (Figure 5D) performed the peptidylcis-trans isomerase activity (PPIase). Peptidylprolylisomerase is a cyclophilin-type peptidyl-prolylcis-trans isomerase, which accelerates the folding of proteins and catalyzes the cis-trans isomerization of prolineimidic peptide bonds in oligopeptides. Protein structure plays a crucial role in its function; if a protein loses its shape at any structural level, it may no longer be functional. Protein secondary structure prediction showed that this protein consists of a mixture of the helix, coils, and strands (β-sheet) distributed throughout the protein sequence (Figure 5E). The β turns (11%), α helix (19%), coils (40%), and extended strands (28%) were observed which showed its stability at the structural level. The amino acid sequences of the PPIase were also analyzed for the putative phosphorylation sites at the NetPhos 3.1 Server (https://services.healthtech.dtu.dk/service.php?NetPhos-3.1, accessed on 18 October 2021). The results showed that the most abundant phosphorylation site is serine residues in PPIase protein sequences (Figure 5F).

#### 3.5.2. PPI Network

Moreover, to understand the mechanism of action of identified PPIase from this study, STRING software was used to predict the interacting partners, and the interaction study showed that the identified protein interacts with other PPIase, GTP binding nuclear proteins, 60S ribosomal protein, and protein kinase. GTP-binding protein is involved in nucleocytoplasmic transport and is required for importing proteins into the nucleus and also for exporting RNA (Figure 6A,B).

They are involved in chromatin condensation and control of the cell cycle. Ribosomal proteins are involved in many processes such as protein synthesis, interaction with the environment, protein with binding function, and protein fate. It shows that PPIase interacts with signaling-related proteins and performs the function for alleviating stress.

## 4. Discussion

Information regarding the pattern of inheritance as well as the nature of gene action involved in resistance against *Sclerotinia sclerotiorum* helps crop geneticists and breeders to elect appropriate selection methods for breeding resistant cultivars. Previous studies have revealed that inheritance of Sclerotinia stem rot resistance varies from crop to crop being monogenic in *Vicia faba* [55] while polygenic in *B. napus* [23,32]. Our study is perhaps the first to report on the genetic investigation and protein expression patterns of Sclerotinia stem rot resistance in *B. juncea*. The meteorological data given in Appendix A revealed that environmental conditions at the experimental site were quite favorable for pathogen proliferation and disease development [56]. The significance of generations for lesion length development observed in this study reveals that Sclerotinia stem rot resistance is a heritable trait. The increase in mean lesion length of F_1_, F_2_, BC_1_P_1_, and BC_1_P_2_ progenies over resistant parents indicates that genes contributing to resistance were recessive. The partial dominance of susceptibility over resistance was again confirmed when the mean lesion length of the F_1_ generation was compared with their respective mid-parent values. A similar result was also reported by Baswana et al. [57] in cauliflower, while overdominance of susceptibility in *Brassica napus* was delineated by Khan et al. [58,59]. Similar to Khan et al. [59] and Zhao et al. [60], we also observed transgressive segregants for resistance in F_2_ and BC_1_P_1_ generations of both populations. Such transgressive segregants were nearly asymptomatic without any lesion development which may be fixed later in selfing generations and can be utilized as a source of resistance in the future.

The significant individual and joint scaling test for mean lesion length in both populations indicate that resistance/susceptibility did not follow the simple Mendelian pattern of inheritance. This indicates the role of epistasis in the genetic control of resistance/susceptibility. Hence, the authors suggest six parameter models most appropriate to explain the inheritance pattern of resistance in the present study. Among six genetic components, our results signify the role of five genetic components in the genetic control of lesion length development, except additive × dominance (j) type of epistasis, which was non-significant in both populations. Among these genetic components, additive (d) gene effect and dominance × dominance (l) type of epistasis were of higher magnitude in the negative direction while dominance (h) gene effect and additive × additive type of epistasis was having higher magnitudes in the positive direction. According to Mather and Jinks [61], the direction of gene effects controlling particular traits is determined by their associated signs. Therefore, in the present study, the additive (h) gene effect and the dominance × dominance (l) type of epistasis mainly governed resistance while the dominance (h) gene effect and the additive × additive (i) type of epistasis imparted susceptibility to *S. sclerotiorum* in Indian mustard. In both populations, estimates of dominance-by-dominance effects (l) were significant and opposite in sign to those of dominance effects alone (h), indicating the role of duplicate epistasis in resistance expression. Similarly, Khan et al. [59] also detected the role of dominance × dominance type of digenic interaction for cotyledon resistance in *Brassica napus* against *S. sclerotiorum*.

Both populations showed high broad and narrow-sense heritability for lesion length development indicating minimal environmental influence. Therefore, selection for resistance may be effective in Indian mustard because all genetic effects were above 80% according to the high broad-sense heritability for lesion length detected in the present study. Another explanation for high heritability during the present study is that the resistance/susceptibility might be controlled by a few major genes. The observed variation between narrow (h^2^ ns) and broad sense (h^2^ bs) heritability exhibits the involvement of dominance effect in the heredity of resistance. However, the value of h^2^ ns was higher than h^2^bs in C-I, which might be due to counteracting effects of additive and dominance genetic variance. Khan et al. [58] observed moderate broad-sense heritability for leaf resistance against *S. sclerotiorum* in oilseed rape. However, the estimates of the genetic parameters, viz., heritability and genetic advance together, are highly desirable for more accurately predicting the genetic gain under selection. In the present study, we found high heritability along with a moderate genetic advance for lesion length development. This indicates that effective progress towards resistance can be made through the selection of lower lesion length as selection efficiency depends upon the magnitudes of heritability and genetic advancement. As duplicate epistasis was prevalent, the selection of transgressive segregants for resistance is also possible in these populations. Similar to this, moderate to high narrow-sense heritability for resistance against *S. sclerotiorum* has been reported by Baswana et al. [57] in cauliflower and Castano et al. [62] in sunflower.

The Wright estimates of the effective factors/minimum number of genes responsible for resistance/susceptibility ranged from 2.46 to 3.50 with an overall average of 2.98. This result strongly reveals the oligogenic nature of inheritance stating that Sclerotinia stem rot resistance might be controlled by at least three major effect genes. Although, inheritance patterns suggested that there may be few to many minor effect genes involved as well, along with major effect genes. Similar to the present study, Vleugels and Bockstaele [63] also detected three major genes responsible for resistance against *S. trifoliorum* in red clover while Moellers et al. [64] reported both the main gene’s effect and epistatic loci responsible for resistance against *S. sclerotiorum* in soybean. The estimates of the average degree of dominance and potency ratio lay between zero to one, which again revealed the partial dominance nature of genes responsible for lesion length development. The estimate of F value is an indicator of association between additive (D) and dominance (H) genetic variance over all loci controlling the trait under study. The negative F value in the present study indicated the presence of partial dominant genes in susceptible parents; therefore, susceptibility is partially dominant over resistance. The ratio of F/$\sqrt{H\times D}$ was close to zero during the present investigation, which reveals that the magnitude and sign of the genes controlling the character are not equal and hence $\sqrt{H/D}$ is not a good estimator of dominance and only explains average dominance for resistance/susceptibility. The results of the present study suggested the role of both additive and non-additive genetic effects—although the non-additive part is slightly higher. Thus, a breeding approach that could exploit both additive and non-additive gene actions would be appropriate in the present situation. Hence, initial single seed descent till high homozygosity is achieved followed by reciprocal recurrent selection in succeeding generations seems to be the most appropriate breeding procedure for improvement and/or introgression of resistance into a desirable agronomic background. However, Barbetti et al. [65] suggested that stem resistance in oilseed brassica against *S. sclerotiorum* could be race/isolate specific(vertical resistance) or race non-specific (durable resistance), the latter being the most optimal and most effective resistance source for breeding novel cultivars with robust resistance across multiple pathotypes of *S. sclerotiorum*. Although, we screened these populations only against a single isolate (*Hisar isolate*) which is a prevalent virulent isolate/race of *S. sclerotiorum* in CCS HAU, Hisar. However, the resistant parent (RH 1222-28) involved in the present study was found resistant/highly tolerant against multiple isolates of different locations as screened in the previous studies [3,5,33,34].

Moreover, the protein expression study revealed the involvement of peptidyl-prolyl-cis/trans-isomerases (PPIases or immunophilins) in resistance against *S. sclerotiorum*. We observed a rapid accumulation of PPIase with a molecular weight of 18.4 kDa in the resistance genotype during early infection stage (6th day after infection). In contrast, in the susceptible genotypes, it appeared late (12th and 18th day after infection). The present study suggested that this PPIase has a definite role in resistance and implicates the corresponding protein as a biomarker for separating susceptible and resistant genotypes during the early stages of plant development. PPIases are the class of stress-responsive proteins of the immunophilin family. It catalyzes various biological functions such as transcription regulation, protein folding and degradation, signal transduction, and reactive oxygen species (ROS) regulation, cell wall strengthening to cope with stress conditions [66,67,68,69,70]. A plethora of recent reports indicate that PPIases are critical resistance/tolerance factors against various biotic and abiotic stresses in crops [69,71,72,73,74,75]. For example, PPIases trigger both salt tolerance and *P. syringae* pv. tabaci resistance in tobacco. Studies reported that PPIase deletion leads to susceptibility toward *P. syringae* in *Arabidopsis thaliana* and provides resistance against *Plasmodiophora brassicae* in *Brassica oleracea* [76]; drought and salt tolerance in sorghum [66,67]; drought tolerance in rice [77]; against fungal pathogen *Leptosphaeria maculans* in *Brassica carinata*-derived *Brassica napus* introgression lines [78]; against *Xanthomonas campestris* in *A. thaliana* [73]. PPIases are involved in a wide range of molecular pathways and play an essential role in protein folding, reactivation of denatured proteins, and restoration of polypeptide active structures. Besides their unique functions, PPIases are part of large chaperone complexes, transmembrane channels responsible for Ca^2+^, and other ion transport events. PPIases have a role in various cellular processes such as signal transduction, RNA processing, protein secretion, cell cycle control, development regulation, photosynthesis, and host–pathogen interactions [79].

It is reported that PPIase has a role in host–pathogen interactions by several mechanisms: (1) Modification of transmembrane and secreted proteins by PPIases of the pathogen to escape or overcome the immune response in intra- or intercellular space of host (most bacteria); (2) stabilization and modification of proteins essential for pathogenesis by PPIases of the host (viruses, protozoa, and some bacteria); (3) suppression of pathogen growth by host PPIases. A study reported that cyclophilin C-CyP (PPIase) with a molecular weight of about 20 kDa isolated from Chinese cabbage (*B. campestris* ssp. *pekinensis* L.) possesses fungistatic activity. In vitro growth of several fungi, including *Candida albicans*, *Rhizoctonia solani*, *Botrytis cinerea*, *Trichoderma harzianum*, and *T. viride* was suppressed by C-CyP [80]. Insilico analysis from this study showed that various signaling proteins are involved in pathogen resistance.

Previous studies indicated that pathogen-mediated host intracellular acidification increased the expression of PPIase. *Sclerotinia sclerotiorum* releases oxalic acid inside the invaded host to cause disease in plants. Oxalic acid is a chief virulence/factor without oxalic acid, the pathogen loses its pathogenicity and becomes non-pathogenic/avirulent. Oxalic acid creates an acidic environment inside the invaded host tissue to cause direct toxicity to living cells, suppresses the host antioxidant defense system, consequently escalating the activities of various cell wall-degrading enzymes to disturb host cell wall integrity [81]. To overcome the adverse effects of oxalic acids, PPIase may act as intracellular pH homeostatic machinery and refold several stress-related proteins to activate H+ extrusion and restore intra-cellular pH of the *S. sclerotiorum* invaded host tissue [82].

Therefore, the present study, apart from being an initial step for further investigation towards the molecular basis of Sclerotinia stem rot resistance, could be beneficial for designing operative breeding programs that might lead to Indian mustard cultivars resistant to this economically important disease.

## Figures and Tables

**Figure 1 genes-12-01784-f001:**
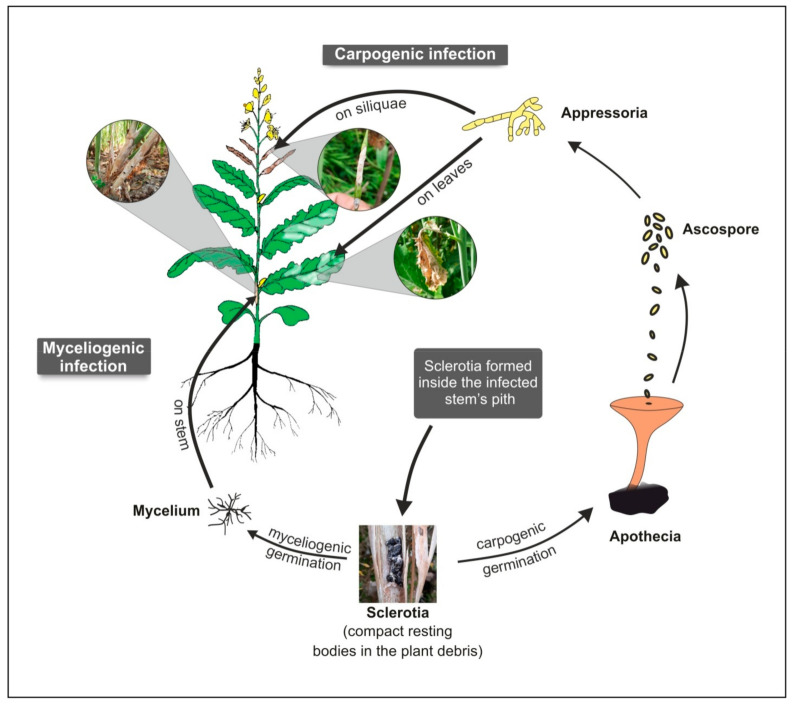
*Sclerotinia sclerotiorum* life cycle and dual infection modes *viz*., myceliogenic and carpogenic means, in Indian mustard.

**Figure 2 genes-12-01784-f002:**
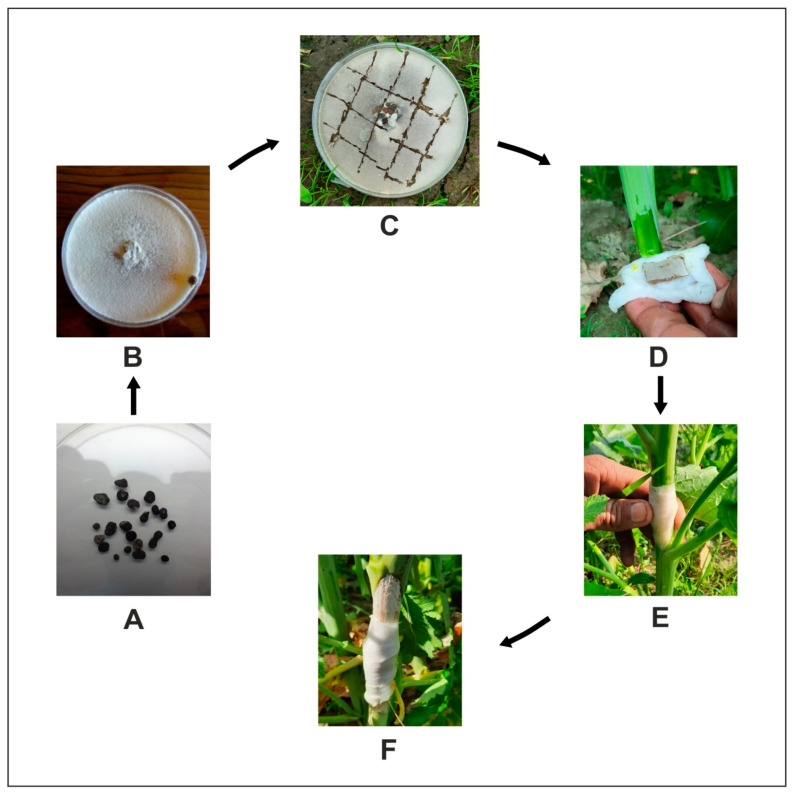
*Sclerotinia sclerotiorum* pure culture preparation and artificial stem inoculation; (**A**) sclerotial samples were surface sterilized with 0.1% mercuric chlorite solution; (**B**) 5-days-old pure culture of S. sclerotiorum; (**C**) 5 mm mycelial bits cuts from 5-days pure culture of *S. sclerotiorum*; (**D**) single mycelial bit along with cotton swab soaked in sterilized distilled water; (**E**) wrapping the parafilm strip around the stem; (**F**) inoculated plant showed characteristics symptoms of water-soaked lesion.

**Figure 3 genes-12-01784-f003:**
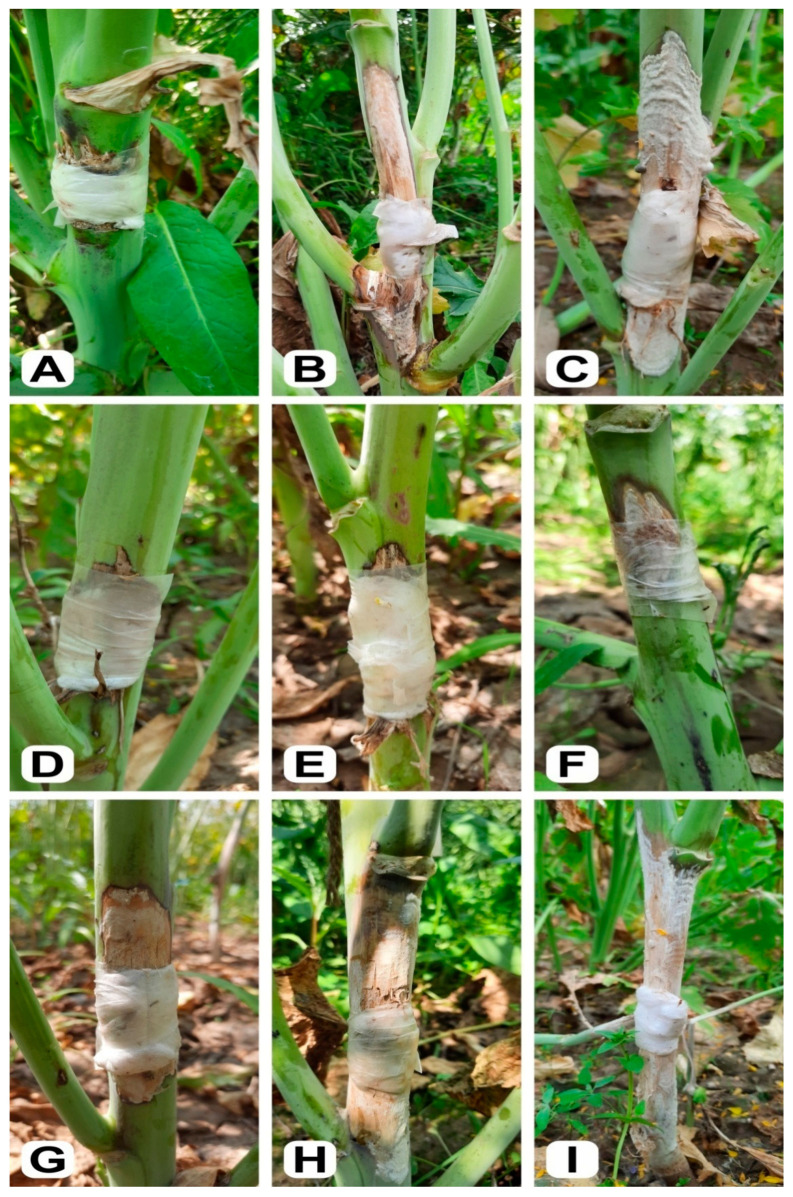
*Sclerotinia sclerotiorum* response exhibited by parental genotypes. (**A**) Resistant parent RH 1222-28 with lesion length <4.0 cm; (**B**) susceptible parents EC 766300 and (**C**) EC 766123 with lesion length > 9.0 cm; disease response in F_2_ plants; (**D**,**E**) highly resistant with lesion length < 2.5 cm; (**F**) resistant response with lesion length between 2.6 to 5.0 cm; (**G**) moderately resistant response with lesion length between 5.0 to 7.5 cm; (**H**) susceptible response with lesion length between 7.6 to 10.0 cm; (**I**) highly susceptible response with lesion length > 10.0 cm.

**Figure 4 genes-12-01784-f004:**
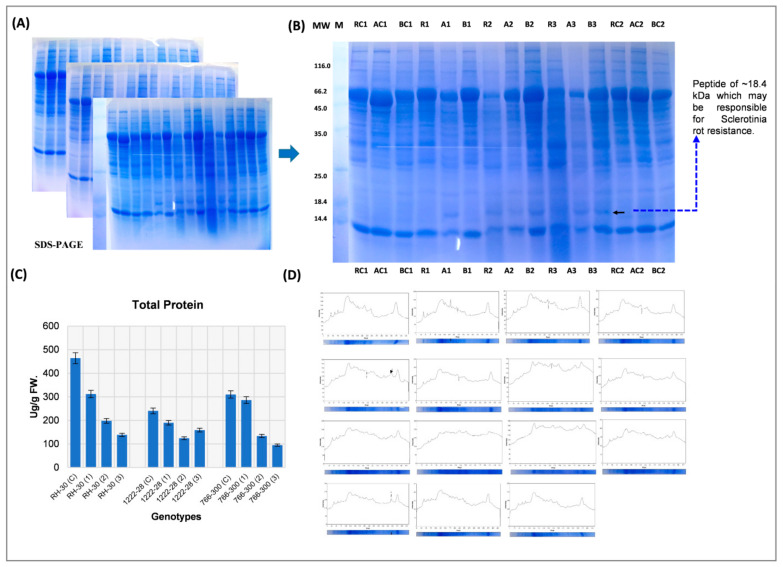
Protein expression profile of various genotypes by using SDS-PAGE: R = RH-30; A = RH 1222-28; B = EC 766,300; C1 and C2 = Control before infection and control at the time of infection; R1, A1 and B1 = First Infection; R2, A2 and B2 = Second Infection; R3, A3 and B3 = Third Infection. Arrow indicating the presence of a particular peptide of ~18 kDa which may be responsible for Sclerotinia rot resistance; (**A**,**B**) total protein content; (**C**) densitometry analysis of expression profile (**D**).

**Figure 5 genes-12-01784-f005:**
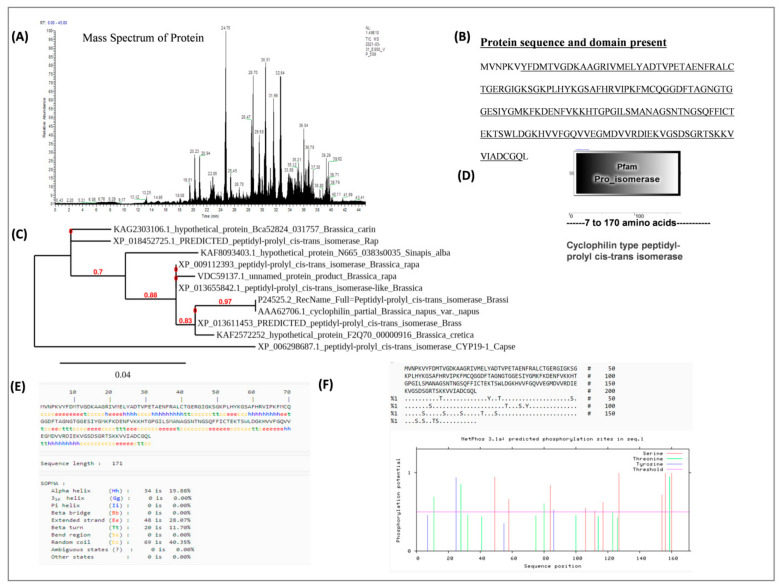
Physicochemical property of PPIase: (**A**) Mass spectrum of protein obtained by sequencing of protein (**B**); amino acid composition of protein (**C**); phylogenetic analysis: Multiple sequence alignment and phylogenetic tree of the deduced amino acid sequence of the PPIase protein with the homologous proteins from other plant species including Brassica. Both the analysis performed using clustal W software with default parameters. (**D**,**E**) Domain analysis and secondary structure prediction andphosphorylation sites in protein structure (**F**).

**Figure 6 genes-12-01784-f006:**
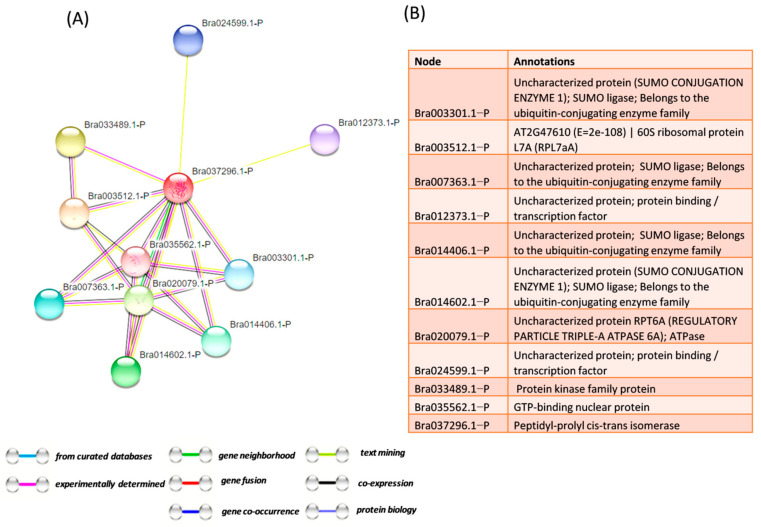
Interaction study using STRING software: (**A**) Protein-Protein Interaction network using default parameters and (**B**) Interacting proteins with annotation. Stronger associations are represented by thick line.

**Table 1 genes-12-01784-t001:** Description of six generations of two crosses, viz., C-1 (RH 1222-28 × EC 766300) and C-II (RH 1222-28 × EC 766123).

Generations	C-I	C-II
P_1_	RH 1222-28	RH 1222-28
P_2_	EC 766300	EC 766123
F_1_	RH 1222-28 × EC 766300	RH 1222-28 × EC 766123
F_2_	F_1_ selfed	F_1_ selfed
BC_1_P_1_	(RH 1222-28 × EC 766300) × RH 1222-28	(RH 1222-28 × EC 766123) × RH 1222-28
BC_1_P_2_	(RH 1222-28 × EC 766300) × EC 766300	(RH 1222-28 × EC 766123) × EC 766123

C-I (RH 1222-28 × EC 766300); C-II (RH 1222-28 × EC 766123).

**Table 2 genes-12-01784-t002:** Scale used for screening different populations against *S. sclerotiorum* as suggested by Garg et al. [20].

Scheme	Lesion Length (cm)	Disease Response	Scale
1.	<2.5	Highly resistant	0
2.	2.6–5.0	Resistant	1
3.	5.1–7.5	Moderately resistant	2
4.	7.6–10.0	Susceptible	3
5.	>10.0	Highly susceptible	4

**Table 3 genes-12-01784-t003:** Analysis of variance (ANOVA) for lesion length (cm) development in two populations of Indian mustard.

Source of Variation	df	Mean Squares	
		C-I	C-II
Replications	2	0.48	1.36
Generations	5	78.53 **	56.41 **
Error	10	0.64	0.72

** Significant at *p* ≤ 0.01; C-I (RH 1222-28 × EC 766300); C-II (RH 1222-28 × EC 766123).

**Table 4 genes-12-01784-t004:** Mean (± SE) comparison for lesion length (cm) among different generations of two populations in Indian mustard.

Generation	Population	
	C-I	C-II
P_1_	4.39 ^d^ ± 0.47	4.39 ^c^ ± 0.47
P_2_	17.40 ^a^ ± 0.66	14.69 ^a^ ± 0.63
F_1_	13.43 ^b^ ± 0.38	11.44 ^b^ ± 0.35
F_2_	10.01 ^c^ ± 0.31	9.57 ^bc^ ± 0.28
BC_1_P_1_	9.08 ^c^ ± 0.32	9.33 ^bc^ ± 0.35
BC_1_P_2_	16.06 ^a^ ± 0.42	14.45 ^a^ ± 0.37

Treatments mean in the same column with different letters differ significantly (*p* ≤ 0.05) based on Duncan’s multiple range test (DMRT); C-I (RH 1222-28 × EC 766300) and C-II (RH 1222-28 × EC 766123).

**Table 5 genes-12-01784-t005:** Estimates of individual and joint scaling tests in two populations of Indian mustard.

Population	Individual Scaling Tests				Joint Scaling Test
	A	B	C	D	χ^2^ (df = 3)
**C-I**	0.34 ± 0.87	1.29 ± 1.13	−8.62 ** ± 1.66	−5.13 ** ± 0.81	43.13 **
**C-II**	2.83 * ± 0.92	2.78 * ± 1.04	−3.66 * ± 1.55	−4.64 ** ± 0.77	38.03 **

** Significant at *p* ≤ 0.01, * Significant at *p* ≤ 0.05 using *t*-test; C-I (RH 1222-28 × EC 766300) and C-II (RH 1222-28 × EC 766123). The A and B scaling tests provided the evidence for the presence of additive × additive (i), additive × dominance (j) and dominance × dominance (l) type gene interactions whereas significance of C and D scaling tests indicated the presence of dominance × dominance and additive × additive component of epistasis.

**Table 6 genes-12-01784-t006:** Estimates of the additive, dominance, and interaction parameters for mean lesion length (cm) in C-I and C-II populations of Indian mustard.

Parameters	Types of Gene Action	Population	
		C-I	C-II
m	Mid parent	10.01 ** ± 0.31	9.57 ** ± 0.28
d	Additive	−6.98 ** ± 0.53	−5.12 ** ± 0.51
h	Dominance	12.78 ** ± 1.72	11.17 ** ± 1.62
i	Additive ×Additive	10.25 ** ± 1.63	9.27 ** ± 1.53
j	Additive × Dominance	−0.48 ± 0.66	0.03 ± 0.65
l	Dominance × Dominance	−11.89 ** ± 2.68	−14.89 ** ± 2.57
Type of epistasis		Duplicate	Duplicate

** Significant at *p* ≤ 0.01 using *t*-test; C-I (RH 1222-28 × EC 766300) and C-II (RH 1222-28 × EC 766123).

**Table 7 genes-12-01784-t007:** Genetic parameters and components of variation for lesion length (cm) in two populations of Indian mustard.

Estimates	Population	
	C-I	C-II
Broad sense heritability (h^2^ bs)	0.84	0.82
Narrow sense heritability (h^2^ ns)	0.86	0.69
Genetic advance (GA)	9.33	8.35
Additive variance (D)	49.76	33.54
Dominance variance (H)	−2.14	12.53
Environmental variance (E)	4.64	4.30
Potence ratio (PR)	0.39	0.37
Average degree of dominance $\sqrt{H/D}$	−0.21	0.61
Covariance between D and H over all loci (F)	−8.99	−1.70
F/$\sqrt{\left(H\times D\right)}$	0.08	0.00
Effective factors/minimum number of genes	3.50	2.46

C-I (RH 1222-28 × EC 766300); C-II (RH 1222-28 × EC 766123).

## Data Availability

Data will be available on reasonable request.

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
