# Peer review of "Genetic and Proteomic Basis of Sclerotinia Stem Rot Resistance in Indian Mustard [Brassica juncea (L.) Czern & Coss.]"

_genes, 2021, doi:10.3390/genes12111784_

Round 1
Reviewer 1 Report
Creation of the cultivars, resistant to the diseases is one of the main problems of breeding. Solving of this problem requires deep knowledge of resistance mechanisms and inheritance. From this point of view the presented article is complex one. It includes phenotyping of of hybrids segregation and and detection of the proteins, involved in the resistance mechanisms.
At the same time I have a comment about table 5. It is not described what are A, B, C, D. Also it is not evident what does this method test?
Author Response
Response to Reviewer reports
Manuscript ID: genes-1447990: “Genetic and proteomic basis of Sclerotinia stem rot resistance in Indian mustard [Brassica juncea (L.).”
We are thankful to the Editor and the reviewers for their keen observations and valuable suggestions regarding the manuscript. We have now revised the manuscript as per the suggestions and point-by-point response to the queries is appended below.
Reviewer #1
Comment 1: Creation of the cultivars, resistant to the diseases is one of the main problems of breeding. Solving of this problem requires deep knowledge of resistance mechanisms and inheritance. From this point of view, the presented article is complex one. It includes phenotyping of hybrids segregation and detection of the proteins, involved in the resistance mechanisms.
At the same time, I have a comment about table 5. It is not described what are A, B, C, D. Also it is not evident what does this method test?
Response:
We are thankful to the learned reviewer for critically evaluation of MS. The means of the different generations were utilized for obtaining the various genetic effects. To predict genetically control of traits in the beginning only additive (d) and dominance (h) genetic effects are assumed to be present. The data were first tested to fit in simple additive-dominance model and presence of epistasis. In table 5, the adequacy of simple additive-dominance model was tested by using A, B, C and D scales. The A and B scaling tests provided the evidence for the presence of additive × additive (i), additive × dominance (j) and dominance × dominance (l) type gene interactions whereas significance of C and D scaling tests indicated the presence of dominance × dominance and additive × additive component of epistasis. When any one of the four scales was found to deviate significantly from zero, the additive–dominance model was considered inadequate. In such case, the joint scale test was employed. Overall, the individual scaling tests (A, B, C, D) and joint scaling test detect the presence and/or absence of non-allelic interactions (epistasis) and test the adequacy of additive-dominance model (Kearsey and Pooni, 1996).

Reviewer 2 Report
This manuscript reports the genetic and proteomic basis of Sclerotinia stem rot resistance in Indian mustard [Brassica juncea (L.) Czern & Coss.]. The genetic pattern of Sclerotinia stem rot resistance in Indian mustard was studied using six generations (P1, P2, F1, F2, BC1P1 & BC1P2) developed from the crossing of one resistant (RH 1222-28) and two susceptible (EC 766300 & EC 766123) genotypes. Genetic analysis revealed that resistance was governed by duplicate epistasis. Comparative proteome analysis between resistant and susceptible genotypes indicated that Peptidyl-prolyl-cis cis-trans isomerase (A0A078IDN6 PPIase) showed high expression in resistance genotype at early infection stage while its expression was delayed in susceptible genotypes.
The work has the potential to be informative. However, some results require further clarification and the work also shows flaws and mistakes in my opinion. I highlight below some comments and major issues.
- In each replication of both of the crosses, 5 plants among the parents, 10 from F1, 100 from F2 generation, 40 from BC1P1 and BC1P2 generations were tagged for artificial stem inoculation. In my opinion, materials are inadequate to support the results.
- In this manuscript, the peptide of ~18.4 kDa may be responsible for Sclerotinia rot resistance. However, I think this method is not convictive to understand the factors/proteins/genes responsible for variation in stem rot-resistance in Brassica. In addition, 18.4 kDa represents many peptides, I am confused about how the Peptidyl-prolyl cis-trans isomerase was identified as the candidate protein.
- The candidate protein PPIase should conduct a more comprehensive analysis including phylogenetic analysis, domain analysis, relative expression level, and function.
- Minor concerns:
-“factors/proteins/gene responsible for variation” should be “factors/proteins/genes responsible for variation”.
-“20ºC” should be”℃”.
-“1mMPMSF” should be” 1mM PMSF”.
Author Response
Response to Reviewer reports
Manuscript ID: genes-1447990: “Genetic and proteomic basis of Sclerotinia stem rot resistance in Indian mustard [Brassica juncea (L.).”
We are thankful to the Editor and the reviewers for their keen observations and valuable suggestions regarding the manuscript. We have now revised the manuscript as per the suggestions and point-by-point response to the queries is appended below.
Reviewer #2
Comment 1: In each replication of both of the crosses, 5 plants among the parents, 10 from F1, 100 from F2 generation, 40 from BC1P1 and BC1P2 generations were tagged for artificial stem inoculation. In my opinion, materials are inadequate to support the results.
Response:
We are thankful to the reviewer for critically evaluation of MS. We have thoroughly revised the manuscript keeping in view your valuable suggestions. We do agree that the sample size may be small in the present study. Since the non-segregating generations represent the homogeneous population while the segregating generations represent the heterogeneous population. The small sample size was used for generations whose variability is only from the environmental origin (homogeneous generations. i.e., Parents and F1), while the large sample size was used for generations whose variability was both environmental and genetic origin (heterogeneous generations, i.e., BCs and F2). Your suggestion on large sample size is praiseworthy and will include in further experiments.
Comment 2: In this manuscript, the peptide of ~18.4 kDa may be responsible for Sclerotinia rot resistance. However, I think this method is not convictive to understand the factors/proteins/genes responsible for variation in stem rot-resistance in Brassica. In addition, 18.4 kDa represents many peptides, I am confused about how the Peptidyl-prolyl cis-trans isomerase was identified as the candidate protein.
Response: We are thankful for keen observation. We have incorporated the material method for sequencing of protein spot in the MS (2.4.2. Experimental procedure for protein sequencing; PP 7). The molecular weight of the sample proteins was determined by using standard protein molecular weight markers which were run simultaneously during SDS-PAGE electrophoresis of the sample protein. The molecular weight of the unknown polypeptides was determined from their Rf value as mentioned below:
Rf=(Distance migrated by proteins)/(Distance migrated by dye)
Further, protein band was cut from CBB stained gel and sequencing of selected protein was done using Ultra-High Resolution Nano-LC MS/MS with 10-15 Cm C18 Column, Q Exactive/Lumos Series Orbitrap Based Mass Spectrometer for identification of protein and molecular weight based on amino acid sequences.
Protein band was cut from CBB stained gel and sequenced using Ultra-High Resolution Nano-LC MS/MS with 10-15 Cm C18 Column, Q Exactive/Lumos Series Orbitrap Based Mass Spectrometer. We have also incorporated a supplementary file for reference of protein sequencing (Supplementary file 1).
Comment 3: The candidate protein PPIase should conduct a more comprehensive analysis including phylogenetic analysis, domain analysis, relative expression level, and function.
Response: We do agree with your kind suggestions for the improvement of MS. However, we have included the phylogenetic and domain analysis of PPIase please see figure 5 (A-F) for reference. Please see the incorporation in text (3.5.1). Your suggestions about the relative expression level and functional analysis of the concerned protein is praiseworthy and will include in future experiments.
Minor concerns:
Comment 4: -“factors/proteins/gene responsible for variation” should be “factors/proteins/genes responsible for variation”.
Response: Changes incorporated in the manuscript as per reviewer suggestions.
Comment 5: -“20ºC” should be “ ℃”.
Response: Changes incorporated in the manuscript as per reviewer suggestions.
Comment 6: -“1mMPMSF” should be” 1mM PMSF”.
Response: Changes incorporated in the manuscript as per reviewer suggestions.

Round 2
Reviewer 2 Report
It has addressed my concerns in the first review.